# *MPG* and *NPRL3* Polymorphisms Are Associated with Ischemic Stroke Susceptibility and Post-Stroke Mortality

**DOI:** 10.3390/diagnostics10110947

**Published:** 2020-11-13

**Authors:** Chang Soo Ryu, Jinkun Bae, In Jai Kim, Jinkwon Kim, Seung Hun Oh, Ok Joon Kim, Nam Keun Kim

**Affiliations:** 1Department of Biomedical Science, College of Life Science, CHA University, Seongnam 13488, Korea; regis2040@nate.com; 2Department of Emergency Medicine, CHA Bundang Medical Center, School of Medicine, CHA University, Seongnam 13496, Korea; galen97@chamc.co.kr; 3Department of Internal Medicine, CHA Bundang Medical Center, School of Medicine, CHA University, Seongnam 13496, Korea; injaikim@cha.ac.kr; 4Department of Neurology, CHA Bundang Medical Center, School of Medicine, CHA University, Seongnam 13496, Korea; kimjinkwon@cha.ac.kr (J.K.); ohsh72@chamc.co.kr (S.H.O.)

**Keywords:** ischemic stroke, DNA repair-related gene, mTOR pathway-regulating gene, post-stroke mortality, polymorphism

## Abstract

Ischemic stroke is a complicated disease which is affected by environmental factors and genetic factors. In this field, various studies using whole-exome sequencing (WES) have focused on novel and linkage variants in diverse diseases. Thus, we have investigated the various novel variants, which focused on their linkages to each other, in ischemic stroke. Specifically, we analyzed the N-methylpurine DNA glycosylase (*MPG*) gene, which plays an initiating role in DNA repair, and the nitrogen permease regulator-like 3 (*NPRL3*) gene, which is involved in regulating the mammalian target of rapamycin pathway. We took blood samples of 519 ischemic stroke patients and 417 controls. Genetic polymorphisms were detected by polymerase chain reaction (PCR), real-time PCR, and restriction fragment length polymorphism (RFLP) analysis. We found that two *NPRL3* polymorphisms (rs2541618 C>T and rs75187722 G>A), as well as the *MPG* rs2562162 C>T polymorphism, were significantly associated with ischemic stroke. In Cox proportional hazard regression models, the *MPG* rs2562162 was associated with the survival of small-vessel disease patients in ischemic stroke. Our study showed that *NPRL3* and *MPG* polymorphisms are associated with ischemic stroke prevalence and ischemic stroke survival. Taken together, these findings suggest that *NPRL3* and *MPG* genotypes may be useful clinical biomarkers for ischemic stroke development and prognosis.

## 1. Introduction

In recent years, the advancement of whole-exome sequencing (WES) has allowed for enhanced screening of genetic diseases [1] and has become a useful tool for studying various diseases [2]. Specifically, the use of WES has resulted in large cohorts of subjects with neurodevelopmental disorders, from which numerous gene mutations have been identified, as well as the confirmation of a role for the identified variants in related pathways of more complex diseases [3,4]. Using this approach, many genes underlying monogenic disorders have now been established; however, the identification of genes involved in multi-genic diseases is more complicated [5]. Furthermore, various studies using WES have identified and focused on the novel, linkage, and haplotype variants in diverse diseases [6,7,8,9,10]. Thus, we have investigated the various novel single-nucleotide polymorphisms, which focused adjacent to one another, in ischemic stroke.

Ischemic stroke is a multifactorial disease that is affected by both genetic [11,12] and environmental factors [13], including advanced age [14], diabetes mellitus [15], family or personal history of stroke [16], high cholesterol [17], smoking [18], hypertension [19], hyperlipidemia [20], and metabolic syndrome [21]. Of these, the primary contributors to ischemic stroke are diabetes mellitus, hypertension, smoking, and hyperlipidemia [22]. Ischemic strokes occur when an artery supplying blood to the brain becomes blocked. This occlusion is often the result of thrombosis, which is responsible for 53% of ischemic strokes, or an embolism, which is responsible for 31% of ischemic strokes [23].

Platelets are a component of the blood responsible for clotting and are directly involved in both thrombus composition and thrombosis [24]. Importantly, many studies have demonstrated a clear association between platelet levels and ischemic stroke [25]. Platelets are formed within the cytoplasm of large precursor cells called megakaryocytes, which reside in the bone marrow [26]. The proliferation of megakaryocyte progenitors is regulated by the mammalian target of rapamycin (mTOR) pathway, which plays an important role in megakaryocyte terminal differentiation, including platelet functions [27]. In the brain, the mTOR pathway regulates numerous processes, including the development of nerve cells and their plasticity over time [28]. Additionally, increased mTOR activity has been shown to facilitate brain recovery after stroke [29].

DNA repair mechanisms also play an important role in the mammalian brain, and polymorphisms of DNA repair-related genes are associated with ischemic stroke susceptibility and short-term recovery [30,31]. In addition, preservation of normal brain function after endogenous damage and protection from brain-related diseases are associated with successful DNA repair, such as base-excision repair [32]. DNA glycosylases are enzymes involved in base-excision repair and are found in many species, including bacteria, yeast, plants, rodents, and humans. As such, there exist multiple subfamilies of these enzymes, including the human alkyladenine DNA glycosylase [33]. Notably, this subfamily of monofunctional glycosylases is involved in the recognition of base lesions, including alterations in purines, and in the initiation of the base-excision repair pathway [34]. Together, these findings support our efforts to assess whether mTOR pathway-related and DNA repair-related genes are associated with ischemic stroke prevalence.

Therefore, in this study, we investigated gene polymorphisms associated with ischemic stroke using WES. Specifically, we investigated the *MPG* gene, which encodes for N-methylpurine DNA glycosylase [35], and the *NPRL3* gene, which encodes for nitrogen permease regulator-like 3, a component of the GATOR1 complex (Appendix A). Importantly, the GATOR1 complex is a regulator of the mTOR pathway [36], and its actions can be blocked by inhibiting mTOR complex 1 activity [37]. As a result of our WES screening, we discovered novel single-nucleotide polymorphisms for *NPRL3* (rs2541618 C>T, rs75187722 G>A) and *MPG* (rs2562162 C>T, rs710079 C>T) that were adjacent to one another in chromosome 16. Thus, we designed a case-control study to investigate the association between the *NPRL3* and *MPG* polymorphisms and ischemic stroke in a Korean population. To the best of our knowledge, this study presents the first piece of evidence of a role for *NPRL3* and *MPG* polymorphisms in ischemic stroke risk in Korean individuals.

## 2. Materials and Methods 

### 2.1. Study Participants

Blood samples were collected from 519 ischemic stroke patients and 417 controls. All study protocols were reviewed and approved by the Institutional Review Board of CHA Bundang Medical Center and followed the recommendations of the Declaration of Helsinki (IRB No. 2013-09-073, date of approval: 6 November 2013). All patients and controls were recruited from the Department of Neurology of CHA Bundang Medical Center, CHA University between 2000 and 2008. This study was approved by the Institutional Review Board of CHA Bundang Medical Center in June 2000 (IRB No. 2013-09-073, date of approval: 6 November 2013) and both informed and written consent was obtained from study participants. 

Ischemic stroke was defined as a stroke (a clinical syndrome characterized by rapidly developing clinical symptoms and signs of the focal or global loss of brain function) with evidence of cerebral infarction in clinically relevant areas of the brain according to magnetic resonance imaging (MRI) scan findings. On the basis of the clinical symptoms and neuroimaging data, two neurologists categorized whole ischemic strokes into four causative subtypes using the criteria of the Trial of Org 10172 in Acute Stroke Treatment (TOAST) [38] as follows: (1) large-artery disease (LAD) defined by an infarction lesion of ≥15 mm in diameter by MRI and significant (>50%) stenosis of the main brain artery or cerebral cortex artery by cerebral angiography, (2) small-vessel disease (SVD) defined by an infarction lesion of <15 and ≥5 mm in diameter by MRI and classic lacunar syndrome without evidence of cerebral cortical dysfunction or detectable cardiac embolism, (3) cardioembolism (CE) defined by arterial occlusions and due to an embolus arising in the heart, as detected by cardiac evaluation, and (4) pathogenesis in which the cause of the stroke was not determined or had more than two causes.

Based on these criteria, 39.9% (n = 207) of patients in the stroke group had LAD, 28.7% (n = 149) had SVD, 10.2% (n = 53) experienced a CE, and 21.2% (n = 110) were of an undetermined pathogenesis. In addition, we selected 417 controls that were matched for sex ratio and age within 5 years in accordance with the patient group. Controls were drawn from subjects visiting our hospitals during the same period for health examinations, including biochemical testing, electrocardiograms, and brain MRIs.

Hypertension was defined as a systolic pressure of >140 mmHg and a diastolic pressure of >90 mmHg on more than one occasion and included patients currently taking hypertensive medications. Diabetes mellitus was defined as fasting plasma glucose levels of >126 mg/dL (7.0 mmol/L) and included patients currently taking diabetic medications. Smoking referred to patients that were current smokers. Hyperlipidemia was defined as a high fasting serum total cholesterol level (≥240 mg/dL) or a history of taking an antihyperlipidemic agent treatment. Patients that possessed more than three of the following five risk factors were diagnosed with metabolic syndrome (MetS): body mass index (BMI) of ≥25.0 kg/m^2^; triglycerides of ≥150 mg/dL; high-density lipoprotein cholesterol (HDL-c) of ≤40 mg/dL in men or ≤50 mg/dL in women; blood pressure ≥140/90 mmHg or patients currently taking hypertensive medication; and fasting plasma glucose of ≥110 mg/dL or patients currently taking insulin or hypoglycemic medication.

### 2.2. Whole-Exome Sequencing (WES) Analysis

The *NPRL3* and *MPG* polymorphisms were selected using WES screening because our purpose was to investigate the various novel single-nucleotide polymorphisms (SNPs), which focused adjacent to one another, in ischemic stroke. After the gene selection progressing, we progressed the polymerase chain reaction (PCR) and restriction fragment length polymorphism (RFLP) because we needed to confirm that the WES data were correct. The significant SNP list was filtered by the significant criteria satisfying *P*-value < 0.05 for Fisher’s exact test, and these significant SNPs were shown through a Manhattan plot (Appendix A). Ten variants were selected by Fisher’s exact test (*P* = upper 0.2%), and we chose four sites in two genes, which were adjacent to one another, that contained a polymorphic genotype as well as one previously reported site (Appendix A, Appendix A). Paired-end sequences produced by a HiSeq Instrument were first mapped on the human genome using the mapping program “BWA” (version 0.7.12). Based on the output BAM file, variant genotyping of each sample was performed with the Haplotype Caller of GATK. Then, an in-house program and SnpEff were applied to filter additional databases including ESP6500, ClinVar, and dbNSFP2.9. For the advanced analysis, we gathered all the per-sample genomic variant call formats (GVCFs) and passed them together to the joint genotyping tool and genotype GVCFs. We calculated genotype frequencies of each individual polymorphism and evaluated the Hardy–Weinberg equilibrium (HWE) to check the data quality and genotype error. 

The association between the case-control status and each individual single-nucleotide polymorphism was analyzed by Fisher’s exact test, which assumed that a rare allele variant had an effect for each polymorphism. Gene enrichment and functional annotation analysis for significant SNPs were performed based on the Gene Ontology database (Gene Ontology unifying biology, available online: https://geneontology.org, accessed on 13 November 2020). The statistical analysis for association test was conducted using PLINK 1.07 (massgeneralbrigham, available online: https://pngu.mgh.harvard.edu/~purcell/plink/, accessed on 13 November 2020).

### 2.3. Genotyping

Genomic DNA from the stroke patients and controls was extracted in blood leukocytes using a G-DEX(TM) on blood extraction kit (Intron Biotechnology, Seongnam, South Korea). Most of the genetic polymorphisms were confirmed by PCR and RFLP. One genetic polymorphism was detected by real-time PCR. The *NPRL3* rs2541618 C>T, *NPRL3* rs75187722 G>A, and *MPG* rs710079 C>T polymorphisms were confirmed by restriction enzyme activation of *Sau*96I, *Hph*I, and *Bcc*I (New England Bio Laboratories, Ipswich, MA, USA) at 37 °C over a period of 16–24 hrs (Appendix A). Each polymorphism genotype was confirmed by electrophoretic separation on 4% agarose gels. In addition, for each polymorphism, 30% of the PCR data were randomly selected for a second PCR assay and were checked using DNA sequencing to validate real-time PCR and RFLP findings.

### 2.4. Post-Stroke Mortality

To estimate the association between *NPRL3*, *MPG* gene polymorphisms, and long-term prognosis after ischemic stroke, the time from stroke occurrence to death was recorded. The death dates of each stroke patient (n = 519) were confirmed using death certificates obtained from the Korean National Statistical Office. The survival statistics were derived from the survival data from 2008 to 2013, and patients who were alive on 31 December 2013 were censored.

### 2.5. Statistical Analysis

Differences in the frequencies of the identified polymorphisms between stroke patients and control subjects were analyzed using Fisher’s exact test and logistic regression. The odds ratio (OR) and 95% confidence interval (CI) were utilized to measure the association between genotype frequencies and stroke. To evaluate the relationship between each specific polymorphism, as well as a combination of the allele, the OR and 95% CI were used. The relationships between polymorphisms and stroke prevalence were calculated using adjusted ORs (AORs) and 95% CIs in logistic regression with adjusted factors such as sex, age, diabetes mellitus, hypertension, hyperlipidemia, and smoking. Statistical significance was accepted at a degree of *P* < 0.05. The false discovery rate (FDR) correction was performed to adjust multiple comparisons. All of the polymorphisms were consistent with the HWE (*P* > 0.05). To estimate ischemic stroke risk, we used three genetic susceptibility models: additive, dominant, and recessive. All *NPRL3* and *MPG* genotypes were converted into numeric values for logistic regression in accordance with their genotypes. Wild homozygotes were designated as “0” in all models. Heterozygotes were designated as “1” in additive and dominant models and “0” in the recessive model. Mutant homozygotes were designated as “1” in dominant and recessive models and “2” in the additive model.

The haplotypes of whole polymorphisms were confirmed using multifactor dimensionality reduction (MDR) to identify combinations with strong synergistic effects. Furthermore, the HAPSTAT program (version 3.0; www.bios.unc.edu/~lin/hapstat/; University of North Carolina, Chapel Hill, NC, USA) was used to estimate the frequency of all haplotypes and confirmed the combinations with strong synergistic effects. Furthermore, the statistical power of positive associations was calculated using G*POWER 3.0 (Universität Düsseldorf: gpower, available online: http://www.psychologie.hhu.de/arbeitsgruppen/allgemeine-psychologie-und-arbeitspsychologie/gpower.html, accessed on 13 November 2020). Survival curves were created from the Cox proportional hazards regression, and the log-rank test was used to estimate the importance of differences between groups. Cox regression models were used to analyze the independent prognostic importance markers, and results were adjusted for various factors, including sex, age, diabetes mellitus, hypertension, hyperlipidemia, and smoking. Statistical significance was defined as *P* < 0.05.

## 3. Results

### 3.1. Baseline Characteristics

The demographic characteristics and clinical factors of the ischemic stroke patients and controls are shown in Table 1 and Appendix A. The controls and ischemic stroke patients consisted of 41.5% and 44.1% males, respectively, and the mean ages of the control and stroke groups were 63.05 ± 11.00 and 63.29 ± 11.95 years, respectively. Some significant differences in clinical variables were confirmed between ischemic stroke patients and controls. Metabolic syndrome, hypertension, and diabetes mellitus were more incident in stroke patients, LAD patients, and SVD patients than controls. In addition, HDL-c was lower in stroke patients and LAD patients than controls, and folate was lower in stroke patients, LAD patients, and SVD patients than controls. Furthermore, homocysteine was higher in stroke patients and LAD patients than controls. The vitamin B12, total cholesterol, and triglyceride had significant differences between CE patients and controls.

### 3.2. Comparison of the Frequencies of NPRL3 and MPG Polymorphisms between Patients with Ischemic Stroke, Subtypes, and Controls

We investigated the *NPRL3* rs2541618 C>T, *NPRL3* rs75187722 G>A, *MPG* rs2562162 C>T, and *MPG* rs710079 C>T polymorphisms in patients with ischemic stroke and controls, as well as by stoke subtype (LAD, SVD, and CE). The data are shown in Table 2 and Table 3. We calculated the AOR using logistic regression analyses including adjusted factors such as age, sex, hypertension, diabetes mellitus, smoking, and hyperlipidemia. The frequencies of the identified *NPRL3* and *MPG* polymorphisms agreed with the predictions of the HWE (*P* > 0.05). 

We found numerous differences in the included polymorphisms between the ischemic stroke patients and control groups. For example, our results showed that the *NPRL3* rs2541618 C>T polymorphism was associated with the prevalence of ischemic stroke: the *NPRL3* rs2541618 TT genotype and the dominant model (CC vs. CT+TT) were both associated with ischemic stroke prevalence. Similarly, the *NPRL3* rs75187722 G>A polymorphism was associated with ischemic stroke prevalence, and the *NPRL3* rs75187722 GA genotype and the dominant model (GG vs. GA+AA) were also significantly associated with ischemic stroke prevalence. However, no polymorphisms were associated with ischemic stroke prevalence after the FDR analysis.

In our ischemic stroke subgroup analysis, we found several associations between the *NPRL3* and *MPG* polymorphisms and stroke subtype. SVD was significantly associated with the *NPRL3* rs2541618 C>T polymorphism (TT and additive model: CC vs. CT vs. TT; dominant model: CC vs. CT+TT; recessive model: CC+CT vs. TT), the *NPRL3* rs75187722 G>A polymorphism (GA and additive model: GG vs. GA vs. AA; dominant model: GG vs. GA+AA), and the *MPG* rs2562162 C>T polymorphism (CT and additive model: CC vs. CT vs. TT; dominant model: CC vs. CT+TT). The *NPRL3* gene polymorphisms and *MPG* gene polymorphisms were further assessed using the FDR analysis. Notably, only the *NPRL3* rs2541618 C>T (TT genotype and additive model), the rs75187722 G>A (additive model and dominant model), and the *MPG* rs2562162 C>T (additive model and dominant model) were significantly associated with SVD after FDR analysis. Additionally, *NPRL3* and *MPG* gene polymorphisms were not significantly associated with LAD or CE stroke patients after FDR analysis. The statistical powers of positive associations measured in this study are shown in Appendix A.

### 3.3. Haplotype Analyses of NPRL3 and MPG Polymorphisms between Ischemic Stroke Patients and Controls

We next proceeded to perform haplotype analyses comparing the ischemic stroke patients and controls (Table 4). The following showed significant associations with stroke prevalence (*P* < 0.05): the T-A-C allele combination of the *NPRL3* rs2541618 C>T/rs75187722 G>A/*MPG* rs2562162 C>T polymorphisms, the C-A-C allele combination of the *NPRL3* rs2541618 C>T/rs75187722 G>A/*MPG* rs710079 C>T polymorphisms, and the T-A-C allele combination of the *NPRL3* rs2541618 C>T/rs75187722 G>A/*MPG* rs710079 C>T polymorphisms. In addition, although it was not significant, the T-A allele combination of the *NPRL3* rs2541618 C>T/rs75187722 G>A polymorphisms showed a trend toward increasing ischemic stroke prevalence. Each polymorphism was also evaluated with FDR analysis. Only the C-A-C allele combination of the *NPRL3* rs2541618 C>T/rs75187722 G>A/*MPG* rs710079 C>T polymorphisms and the T-A-C allele combination of the *NPRL3* rs2541618 C>T/rs75187722 G>A/*MPG* rs710079 C>T polymorphisms were significantly associated with stroke prevalence after FDR analysis.

### 3.4. Combined Effects of NPRL3 and MPG Polymorphisms and Clinical Factors

To determine if other factors contributed to the association of genotype and ischemic stroke, we stratified our genotype analyses by clinical variables (Appendix A). Moreover, we clarified the number of each subgroup in stratification analysis. These stratified analyses revealed many combined effects of genotype and ischemic stroke risk factors. Therefore, we next performed an analysis of interactions between the *NPRL3* and *MPG* polymorphisms and environmental factors (Table 5, Appendix A). Moreover, we clarified the number of each subgroup in interaction analysis. We found that the *NPRL3* rs2541618 CT+TT genotype had synergistic effects with hypertension, diabetes mellitus, hyperlipidemia, and smoking (Table 5). Especially, *NPRL3* rs2541618 CT+TT with hypertension (AOR, 3.120; 95% CI, 2.116–4.599) was shown to significantly increase ischemic stroke prevalence. In addition, *NPRL3* rs2541618 CT+TT with diabetes mellitus (AOR, 2.904; 95% CI, 1.758–4.796) was shown to significantly increase ischemic stroke prevalence. Furthermore, *NPRL3* rs2541618 CT+TT had a combinatorial effect with HDL-c<40(M)/50(F) (AOR, 6.364; 95% CI, 4.103–9.869) and *NPRL3* rs2541618 CT+TT was shown to possess a combinatorial effect with folate ≤3.54 nmol/L (AOR, 4.866; 95% CI, 2.527–9.370 (Appendix A). Similarly, we found synergistic effects between this polymorphism and several clinical factors (hyperlipidemia, smoking, high homocysteine levels, and low prothrombin time levels) (Table 5 and Appendix A). Furthermore, we found that the *NPRL3* rs75187722 GA+AA genotype had synergistic effects with low HDL-c levels and high platelet levels (Appendix A). 

Similarly, the *MPG* rs2562162 CT+TT genotype exhibited synergistic effects with hypertension, diabetes mellitus, hyperlipidemia, and smoking (Table 5). Especially, *MPG* rs2562162 CT+TT with hypertension (AOR, 3.256; 95% CI, 2.193–4.833) was shown to significantly increase ischemic stroke prevalence. Moreover, *MPG* rs2562162 CT+TT with diabetes mellitus (AOR, 2.984; 95% CI, 1.759–5.063) was shown to significantly increase ischemic stroke prevalence. Furthermore, *MPG* rs2562162 CT+TT had a combinatorial effect with HDL-c<40(M)/50(F) (AOR, 7.330; 95% CI, 4.564–11.773) and *MPG* rs2562162 CT+TT was shown to possess a combinatorial effect with folate ≤ 3.54 nmol/L (AOR, 5.601; 95% CI, 2.651–11.836) (Appendix A). In addition, the *MPG* rs2562162 CT+TT genotype exhibited synergistic effects with several clinical factors (hyperlipidemia, smoking, and low prothrombin time levels) (Table 5 and Appendix A). Moreover, the *MPG* rs710079 CT+TT genotype had synergistic effects with low HDL-c levels and low folate levels (Appendix A).

### 3.5. Analysis of NPRL3 and MPG Polymorphisms with Respect to Survival in Ischemic Stroke Patients and Subtypes 

Our final analysis investigated the relationships between the *NPRL3* and *MPG* polymorphisms and survival in ischemic stroke patients (Figure 1, Appendix A). To this end, we performed a Cox proportional analysis, which showed that the *NPRL3* rs2541618 C>T, *NPRL3* rs75187722 G>A, *MPG* rs2562162 C>T, and *MPG* rs710079 C>T polymorphisms were not associated with survival in ischemic stroke patients. However, the *MPG* rs2562162 CT genotype and dominant model were associated with survival of SVD patients in the Cox proportional hazard regression models (CC vs. CT and CC vs. CT+TT analyses; Figure 1A, Appendix A). Conversely, the *MPG* rs2562162 CT genotype was not associated with survival in LAD or CE patients in our analyses (Appendix A). 

In addition, we found a significant association between the *MPG* rs2562162 C>T dominant model and SVD patients with hypertension using the Cox proportional hazard regression model (CC vs. CT+TT; Figure 1B). Similarly, the *MPG* rs710079 CT genotype was associated with survival in stroke patients with diabetes mellitus using the Cox proportional hazard regression model (CC vs. CT; Figure 1C), and the *NPRL3* rs2541618 C>T dominant model was associated with survival in stroke patients with hyperlipidemia (CC vs. CT+TT; Figure 1D). Finally, we performed a stepwise Cox regression analysis to confirm the covariant effects. As a result, we found that the *MPG* rs2562162 (in SVD patients), *MPG* rs710079 (with diabetes mellitus), and *NPRL3* rs2541618 (with hyperlipidemia) polymorphisms and age were associated with mortality in stroke patients (Appendix A).

### 3.6. Clinical Factors in Ischemic Stroke Patients, Subtypes Stratified by NPRL3 and MPG Polymorphisms 

Using an analysis of variance (ANOVA), we confirmed that the *NPRL3* rs75187722 AA genotype was associated with increased uric acid levels compared to the uric acid levels with the *NPRL3* rs75187722 GG genotype (GG vs. GA vs. AA; *P* = 0.028) in ischemic stroke patients. Furthermore, the *MPG* rs256262 CT+TT was associated with increased platelet levels compared to the platelet levels with the *MPG* rs256262 CC genotype (CC vs. CT+TT; *P* = 0.022) in ischemic stroke patients (Appendix A). 

Subsequently, we performed an ANOVA based on the individual ischemic stroke subgroups. We found that the *NPRL3* rs2541618 CT+TT was associated with increased platelet levels compared to the platelet levels with the *NPRL3* rs2541618 CC genotype in the LAD patients (CC vs. CT+TT; *P* = 0.030). Moreover, the *NPRL3* rs2541618 CT+TT decreased HDL-c compared to HDL-c with the *NPRL3* rs2541618CC genotype in the LAD patients (CC vs. CT+TT; *P* = 0.038) (Appendix A). With respect to the SVD patients, the *NPRL3* rs2541618 TT genotype increased fibrinogen levels compared to fibrinogen levels with *NPRL3* rs2541618 CC+CT (CC+CT vs. TT; *P* = 0.034) (Appendix A).

## 4. Discussion

In this study, we sought to determine novel markers with potential use in clinical diagnostics. To this end, we selected four polymorphisms (*NPRL3* rs2541618 C>T, *NPRL3* rs75187722 G>A, *MPG* rs2562162 C>T, and *MPG* rs710079 C>T) to investigate with respect to ischemic stroke in a Korean population. 

NPRL3 is a known regulator of mTOR activity that promotes neuronal survival in stroke patients and is associated with focal epilepsy [39,40]. Importantly, increased mTOR activity accelerates brain recovery after stroke, and focal epilepsy is associated with the manifestation of ischemic cerebrovascular disease [29,41]. Consequently, the *NPRL3* gene has been previously linked to ischemic stroke prevalence due to its relationship with cerebrovascular disease. Therefore, we hypothesized that *NPRL3* polymorphisms affect ischemic stroke prevalence and prognosis. In agreement with our hypothesis, our analyses showed that the *NPRL3* rs2541618 C>T and *NPRL3* rs75187722 G>A genotypes were significantly associated with the prevalence of ischemic stroke, and this association was strengthened when limited to ischemic stroke patients with SVD. Interestingly, the *NPRL3* rs2541618 C>T polymorphism was associated with decreased long-term survival after stroke in patients with hyperlipidemia. Although we did not identify the cause of the high mortality of stroke patients with the *NPRL3* rs2541618 CT+TT dominant model, our analysis did show that the *NPRL3* rs2541618 polymorphism was significantly associated with the rate of ischemic stroke patient survival. Furthermore, we confirmed that the *NPRL3* rs2541618 and rs75187722 polymorphisms were significantly associated with fibrinogen and uric acid, which are risk factors of ischemic stroke [42,43]. 

Polymorphisms in *MPG* and other base-excision repair-related genes due to age-related or oxidative DNA damage are major contributors to stroke [44,45,46]. Furthermore, base-excision repair, which is initiated by the *MPG* gene, was reported to promote ischemia-reperfusion injury in the brain [47]. These ischemia-reperfusion injuries alter the blood flow restoration at the post-ischemic tissue and lead to further tissue damage [48]. Therefore, we hypothesized that the *MPG* gene is associated with ischemic stroke prevalence and prognosis. In our analyses, we found that the *MPG* rs2562162 C>T genotype was significantly associated with prevalence of the SVD subtype of ischemic stroke. Interestingly, the *MPG* rs2562162 C>T polymorphism was significantly associated with increased long-term survival of SVD patients with and without hypertension, whereas this same polymorphism was not associated with mortality in LAD or CE patients. Although we did not identify the cause of low mortality in SVD patients (with or without hypertension) with the *MPG* rs2562162 dominant model, our analysis did show that the *MPG* rs2562162 polymorphism was significantly associated with the rate of ischemic stroke patient survival. Furthermore, we confirmed that the *MPG* rs2562162 dominant model was significantly associated with risk factors related to the SVD subtype of ischemic stroke, such as platelet levels [49].

This study has several limitations that must be considered when interpreting our results. First, it is not yet clear whether the gene polymorphisms predicted the phenotypes related to ischemic stroke prevalence. Second, we examined a limited pool of patients for our subgroup analyses. Third, the control group in our study was not completely healthy, as some required medical attention. Therefore, it is difficult to identify a casual effect for the clinical risk factors examined in these groups. Additional studies such as the replication study are needed to confirm that the *NPRL3* and *MPG* genes play a crucial role in ischemic stroke pathogenesis and to provide more evidence that the regulation of *NPRL3* and *MPG* expression or activation can be used as a tool to prevent ischemic stroke. Although expression of these genes has not been studied in ischemic stroke, the association between several risk factors of ischemic stroke, including inflammation activation and these gene polymorphisms, elicited speculation about the biomarker’s possible usefulness in early diagnosis [50,51,52,53]. We hypothesized that it could be affected by the decision of ischemic stroke diagnosis in the combined high-risk factor levels and these polymorphisms. Furthermore, the results of a previous genome-wide association study (GWAS) showed that *NPRL3* rs2541618 and *MPG* rs2562162 were associated with SVD patients and controls in the European population (Appendix A). Therefore, our findings suggest that these polymorphisms are potential biomarkers to diagnose ischemic stroke and assess risk.

Authors should discuss the results and how they can be interpreted from the perspective of previous studies and of the working hypotheses. The findings and their implications should be discussed in the broadest context possible. Future research directions may also be highlighted.

## 5. Conclusions

In conclusion, we found that the *NPRL3* rs2541618 C>T, *NPRL3* rs75187722 G>A, and *MPG* rs2562162 C>T genotypes were all strongly associated with ischemic stroke prevalence. In a further analysis, we found that some of these genotypes were associated with different stroke subtypes; specifically, SVD was associated with *NPRL3* rs2541618 TT genotype, *NPRL3* rs75187722 GA genotype and the dominant model, and *MPG* rs2562162 CT genotypes and dominant model. Finally, we found that the dominant models of *NPRL3* rs2541618, *NPRL3* rs75187722, and *MPG* rs2562162 all showed synergistic effects with clinical risk factors of the SVD subtype and ischemic stroke, including platelet level, fibrinogen, hypertension, and diabetes mellitus. Taken together, these results suggest that *NPRL3* and *MPG* polymorphisms play a role in ischemic stroke. Consequently, we identified associations between the *NPRL3* rs2541618 C>T, *NPRL3* rs75187722 G>A, and *MPG* rs2562162 C>T polymorphisms with the prevalence of ischemic stroke in a Korean population as well as a significant association between *NPRL3* and *MPG* gene polymorphisms and post-stroke mortality. However, the specific mechanisms underlying these effects need to be investigated in future research.

## Figures and Tables

**Figure 1 diagnostics-10-00947-f001:**
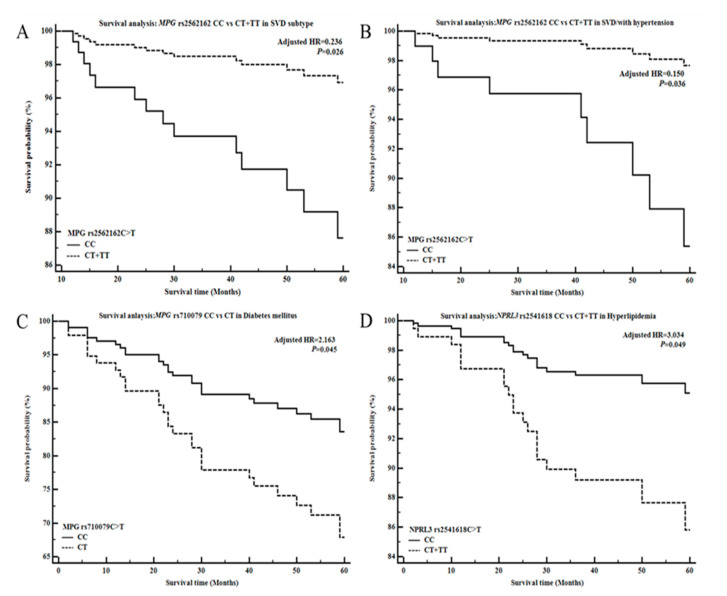
Survival analysis of the small-vessel disease (SVD) in ischemic stroke using the Cox proportional hazards model, according to the *MPG* rs2562162 C>T, *MPG* rs710079 C>T, and *NPRL3* rs2541618 C>T polymorphisms. Survival probability of ischemic stroke patients is shown based on (**A**) the *MPG* rs2562162 CC vs. CT+TT dominant model, (**B**) the *MPG* rs2562162 CC vs. CT+TT dominant model with hypertension, (**C**) the *MPG* rs2562162 CC vs. CT genotypes, and (**D**) the *NPRL3* rs2541618 CC vs. CT+TT dominant model with hyperlipidemia. Note for Figure 1: HR, hazard ratio.

**Table 1 diagnostics-10-00947-t001:** Comparison of baseline characteristics between ischemic stroke patients, ischemic stroke subgroups, and controls.

Characteristics	Controls (n = 417)	Stroke Patients (n = 519)	*P* ^a^	LAD Patients (n = 207)	*P* ^a^	SVD Patients (n = 149)	*P* ^a^	CE Patients (n = 53)	*P* ^a^
Males, n (%)	173 (41.5)	229 (44.1)	0.609	89 (43.0)	0.819	72 (48.3)	0.368	21 (39.6)	0.866
Age, (years, mean ± SD)	63.05 ± 11.00	63.29 ± 11.95	0.747	64.36 ± 11.81	0.172	60.76 ± 11.63	0.032	66.58 ± 12.55	0.031
Smoking, n (%)	140 (33.6)	194 (37.4)	0.405	77 (37.2)	0.535	54 (36.2)	0.682	16 (30.2)	0.724
MetS, n (%)	94 (22.5)	238 (45.9)	**<0.0001**	112 (54.1)	**<0.0001**	68 (45.6)	**0.0001**	20 (37.7)	0.070
HTN, n (%)	171 (41.0)	331 (63.8)	**0.0001**	131 (63.3)	**0.003**	92 (61.7)	**0.011**	30 (56.6)	0.189
DM, n (%)	56 (13.4)	142 (27.4)	**<0.0001**	55 (26.6)	**0.001**	47 (31.5)	**<0.0001**	12 (22.6)	0.132
Hyperlipidemia, n (%)	95 (22.8)	156 (30.1)	0.057	70 (33.8)	0.027	44 (29.5)	0.207	10 (18.9)	0.603

SD, standard deviation; LAD, large artery disease; SVD, small-vessel disease; CE, cardioembolism; MetS, metabolic syndrome; HTN, hypertension; DM, diabetes mellitus. ^a^
*P*-values were calculated by two-sided *t*-test for continuous variables and chi-square test for categorical variables. *P*-values < 0.05 are shown in bold.

**Table 2 diagnostics-10-00947-t002:** Comparison of the *NPRL3* and *MPG* polymorphism frequencies between ischemic stroke patients and controls.

Genotypes	Controls (n = 417)	Stroke Patients (n = 519)	AOR (95% CI) *	*P* ^†^	*P* ^‡^
*NPRL3* rs2541618 C>T					
CC	205 (49.2)	225 (43.4)	1.000 (reference)		
CT	178 (42.7)	238 (45.9)	1.270 (0.944–1.710)	0.115	0.153
TT	34 (8.2)	56 (10.8)	1.679 (1.011–2.787)	**0.045**	0.180
Additive (CC vs. CT vs. TT)			1.205 (0.979–1.485)	0.079	0.161
Dominant (CC vs. CT+TT)			1.333 (1.004–1.770)	**0.047**	0.094
Recessive (CC+CT vs. TT)			1.504 (0.924–2.449)	0.101	0.366
HWE-*P*	0.591	0.555			
*NPRL3* rs75187722 G>A					
GG	333 (79.9)	438 (84.4)	1.000 (reference)		
GA	82 (19.7)	77 (14.8)	0.638 (0.438–0.931)	**0.020**	0.080
AA	2 (0.5)	4 (0.8)	0.979 (0.172–5.586)	0.981	0.981
Additive (GG vs. GA vs. AA)			0.742 (0.531–1.037)	0.081	0.161
Dominant (GG vs. GA+AA)			0.650 (0.448–0.942)	**0.023**	0.092
Recessive (GG+GA vs. AA)			1.060 (0.186–6.039)	0.948	0.948
HWE-*P*	0.197	0.762			
*MPG* rs2562162 C>T					
CC	239 (57.3)	266 (51.3)	1.000 (reference)		
CT	151 (36.2)	217 (41.8)	1.265 (0.950–1.684)	0.107	0.153
TT	27 (6.5)	36 (6.9)	1.213 (0.696–2.114)	0.495	0.661
Additive (CC vs. CT vs. TT)			1.175 (0.944–1.463)	0.149	0.198
Dominant (CC vs. CT+TT)			1.252 (0.954–1.643)	0.106	0.141
Recessive (CC+CT vs. TT)			1.105 (0.642–1.901)	0.719	0.948
HWE-*P*	0.633	0.355			
*MPG* rs710079 C>T					
CC	290 (69.5)	376 (72.4)	1.000 (reference)		
CT	118 (28.3)	136 (26.2)	0.962 (0.709–1.304)	0.801	0.801
TT	9 (2.2)	7 (1.3)	0.500 (0.174–1.436)	0.198	0.396
Additive (CC vs. CT vs. TT)			0.893 (0.680–1.172)	0.413	0.413
Dominant (CC vs. CT+TT)			0.925 (0.686–1.246)	0.606	0.606
Recessive (CC+CT vs. TT)			0.489 (0.170–1.401)	0.183	0.366
HWE-*P*	0.454	0.173			

AOR, adjusted odds ratio; HWE, Hardy–Weinberg equilibrium; 95% CI, 95% confidence interval; NPRL3, nitrogen permease receptor like-3; MPG, N-methylpurine DNA glycosylase. * Adjusted by age, sex, hypertension, diabetes mellitus, hyperlipidemia, and smoking. ^†^
*P*-value calculated by logistic regression analysis; ^‡^
*P*-value calculated by false discovery rate test; *P*-values < 0.05 are shown in bold.

**Table 3 diagnostics-10-00947-t003:** Comparison of the *NPRL3* and *MPG* polymorphism frequencies between ischemic stroke subtype patients and controls.

Genotype	Controls (n = 417)	LAD (n = 207)	AOR (95% CI) *	*P* ^†^	*P* ^‡^	SVD (n = 149)	AOR (95% CI) *	*P* ^†^	*P* ^‡^	CE (n = 53)	AOR (95% CI) *	*P* ^†^	*P* ^‡^
*NPRL3* rs2541618 C>T													
CC	205 (49.2)	93 (44.9)	1.000 (reference)			57 (38.3)	1.000 (reference)			22 (41.5)	1.000 (reference)		
CT	178 (42.7)	89 (43.0)	1.040 (0.717–1.509)	0.835	0.870	73 (49.0)	1.375 (0.902–2.097)	0.138	0.185	27 (50.9)	1.402 (0.766–2.568)	0.274	0.737
TT	34 (8.2)	25 (12.1)	1.860 (1.015–3.408)	**0.045**	0.178	19 (12.8)	2.406 (1.225–4.725)	**0.011**	**0.043**	4 (7.5)	1.069 (0.329–3.480)	0.911	0.911
Additive (CC vs. CT vs. TT)			1.237 (0.946–1.618)	0.120	0.239		1.496 (1.102–2.031)	**0.010**	**0.037**		1.172 (0.747–1.838)	0.490	0.927
Dominant (CC vs. CT+TT)			1.159 (0.815–1.647)	0.411	0.548		1.524 (1.019–2.279)	**0.041**	0.054		1.342 (0.747–2.412)	0.326	0.775
Recessive (CC+CT vs. TT)			1.832 (1.028–3.262)	**0.040**	0.160		2.059 (1.087–3.899)	**0.027**	0.106		0.893 (0.295–2.701)	0.841	0.841
*NPRL3* rs75187722 G>A													
GG	333 (79.9)	179 (86.5)	1.000 (reference)			130 (87.2)	1.000 (reference)			43 (81.1)	1.000 (reference)		
GA	82 (19.7)	28 (13.5)	0.622 (0.380–1.017)	0.058	0.207	18 (12.1)	0.474 (0.261–0.860)	**0.014**	0.056	9 (17.0)	0.835 (0.388–1.797)	0.644	0.737
AA	2 (0.5)	0 (0.0)	NA	0.995	0.995	1 (0.7)	1.301 (0.112–15.114)	0.834	0.834	1 (1.9)	2.829 (0.238–33.596)	0.410	0.911
Additive (GG vs. GA vs. AA)			0.599 (0.370–0.968)	**0.037**	0.146		0.535 (0.306–0.934)	**0.028**	**0.037**		0.969 (0.491–1.914)	0.927	0.927
Dominant (GG vs. GA+AA)			0.606 (0.371–0.989)	**0.045**	0.180		0.495 (0.277–0.886)	**0.018**	**0.044**		0.898 (0.429–1.878)	0.775	0.775
Recessive (GG+GA vs. AA)			NA	0.996	0.996		1.468 (0.127–16.980)	0.759	0.759		2.690 (0.230–31.473)	0.430	0.841
*MPG* rs2562162 C>T													
CC	239 (57.3)	105 (50.7)	1.000 (reference)			65 (43.6)	1.000 (reference)			28 (52.8)	1.000 (reference)		
CT	151 (36.2)	88 (42.5)	1.359 (0.940–1.965)	0.103	0.207	72 (48.3)	1.569 (1.038–2.372)	**0.033**	0.066	20 (37.7)	1.156 (0.621–2.152)	0.647	0.737
TT	27 (6.5)	14 (6.8)	1.189 (0.576–2.452)	0.640	0.853	12 (8.1)	1.708 (0.786–3.708)	0.176	0.343	5 (9.4)	1.276 (0.440–3.697)	0.654	0.911
Additive (CC vs. CT vs. TT)			1.212 (0.916–1.605)	0.179	0.239		1.422 (1.040–1.945)	**0.027**	**0.037**		1.121 (0.718–1.753)	0.615	0.927
Dominant (CC vs. CT+TT)			1.332 (0.937–1.894)	0.110	0.219		1.589 (1.069–2.363)	**0.022**	**0.044**		1.153 (0.643–2.067)	0.633	0.775
Recessive (CC+CT vs. TT)			1.052 (0.522–2.121)	0.886	0.996		1.412 (0.665–2.996)	0.369	0.492		1.179 (0.421–3.300)	0.754	0.841
*MPG* rs710079 C>T													
CC	290 (69.5)	151 (72.9)	1.000 (reference)			115 (77.2)	1.000 (reference)			38 (71.7)	1.000 (reference)		
CT	118 (28.3)	55 (26.6)	0.968 (0.652–1.436)	0.870	0.870	33 (22.1)	0.797 (0.499–1.274)	0.344	0.344	14 (26.4)	0.893 (0.461–1.728)	0.737	0.737
TT	9 (2.2)	1 (0.5)	0.217 (0.026–1.825)	0.160	0.319	1 (0.7)	0.293 (0.035–2.453)	0.257	0.343	1 (1.9)	0.819 (0.098–6.834)	0.854	0.911
Additive (CC vs. CT vs. TT)			0.858 (0.598–1.231)	0.406	0.406		0.738 (0.482–1.129)	0.161	0.161		0.894 (0.501–1.594)	0.704	0.927
Dominant (CC vs. CT+TT)			0.910 (0.616–1.344)	0.636	0.636		0.754 (0.476–1.196)	0.230	0.230		0.891 (0.468–1.695)	0.725	0.775
Recessive (CC+CT vs. TT)			0.211 (0.025–1.760)	0.151	0.301		0.294 (0.035–2.488)	0.261	0.492		0.780 (0.094–6.509)	0.818	0.841

AOR, adjusted odds ratio; HWE, Hardy–Weinberg equilibrium; 95% CI, 95% confidence interval; LAD, large-artery disease; SVD, small-vessel disease; CE, cardioembolism; NA, not applicable; NPRL3, nitrogen permease receptor like-3; MPG, N-methylpurine DNA glycosylase. * Adjusted by age, sex, hypertension, diabetes mellitus, hyperlipidemia, and smoking. ^†^
*P*-value calculated by logistic regression analysis; ^‡^
*P*-value calculated by false discovery rate test; *P*-values < 0.05 are shown in bold.

**Table 4 diagnostics-10-00947-t004:** Haplotype analyses for the *NPRL3* and *MPG* polymorphisms in ischemic stroke patients and controls.

Haplotypes	Controls (2n = 834)	Stroke (2n = 1038)	OR (95% CI)	*P*	*P* *
*NPRL3* rs2541618 C>T/*NPRL3* rs75187722 G>A/*MPG* rs2562162 C>T/*MPG* rs710079 C>T					
C-G-C-C	418 (50.1)	525 (50.6)	1.000 (reference)		
C-A-C-C	21 (2.6)	10 (1.0)	0.379 (0.177–0.814)	**0.010**	0.129
C-A-T-T	4 (0.4)	0 (0.0)	0.088 (0.005–1.649)	**0.039**	0.254
T-A-C-C	2 (0.3)	10 (1.0)	3.981 (0.867–18.270)	0.077	0.335
*NPRL3* rs2541618 C>T/*NPRL3* rs75187722 G>A/*MPG* rs2562162 C>T					
C-G-C	469 (56.2)	568 (54.7)	1.000 (reference)		
C-A-T	9 (1.1)	4 (0.4)	0.367 (0.112–1.200)	0.098	0.196
T-G-T	162 (19.5)	243 (23.4)	1.239 (0.981–1.564)	0.072	0.196
T-A-C	0 (0.0)	7 (0.6)	12.390 (0.705–217.600)	**0.019**	0.112
*NPRL3* rs2541618 C>T/*NPRL3* rs75187722 G>A/*MPG* rs710079 C>T					
C-G-C	446 (53.5)	568 (54.7)	1.000 (reference)		
C-G-T	56 (6.7)	49 (4.8)	0.687 (0.459–1.028)	0.067	0.133
C-A-C	29 (3.5)	10 (1.0)	0.271 (0.131–0.562)	**0.0002**	**0.001**
T-A-C	0 (0.0)	14 (1.4)	22.780 (1.354–383.100)	**0.001**	**0.002**
*NPRL3* rs2541618 C>T/*MPG* rs2562162 C>T/*MPG* rs710079 C>T					
C-C-C	440 (52.8)	532 (51.3)	1.000 (reference)		
T-T-C	159 (19.1)	237 (22.8)	1.233 (0.972–1.563)	0.084	0.586
*NPRL3* rs2541618 C>T/*NPRL3* rs75187722 G>A					
C-G	502 (60.2)	608 (58.6)	1.000 (reference)		
T-A	0 (0.0)	5 (0.5)	9.084 (0.501–164.800)	0.068	0.154
*NPRL3* rs2541618 C>T/*MPG* rs2562162 C>T					
C-C	545 (65.4)	642 (61.9)	1.000 (reference)		
T-T	162 (19.5)	243 (23.4)	1.273 (1.012–1.602)	**0.039**	0.116
*NPRL3* rs75187722 G>A/*MPG* rs2562162 C>T					
G-C	553 (66.3)	669 (64.5)	1.000 (reference)		
G-T	195 (23.4)	284 (27.3)	1.204 (0.972–1.492)	0.090	0.147
A-T	10 (1.2)	5 (0.5)	0.413 (0.140–1.217)	0.098	0.147

OR, odds ratio; 95% CI, 95% confidence interval; NPRL3, nitrogen permease receptor like-3; MPG, N-methylpurine DNA glycosylase. * *P*-value calculated by false discovery rate test. *P*-values < 0.05 are shown in bold.

**Table 5 diagnostics-10-00947-t005:** Ischemic stroke incidence according to an interaction analysis between *NPRL3* and *MPG* genotypes and environmental factors.

Characteristics	*NPRL3* rs2541618 CC	*NPRL3* rs2541618 CT+TT	*NPRL3* rs75187722 GG	*NPRL3* rs75187722 GA+AA	*MPG* rs2562162 CC	*MPG* rs2562162 CT+TT	*MPG* rs710079 CC	*MPG* rs710079 CT+TT
Age (936)								
<63	1.000 (reference)	1.436 (0.951–2.167)	1.000 (reference)	0.589 (0.343–1.014)	1.000 (reference)	1.435 (0.952–2.164)	1.000 (reference)	0.903 (0.573–1.424)
≥63	0.903 (0.594–1.373)	0.996 (0.655–1.515)	0.762 (0.554–1.049)	0.746 (0.443–1.255)	0.866 (0.583–1.288)	1.086 (0.712–1.656)	0.785 (0.558–1.104)	0.778 (0.501–1.207)
Sex (936)								
Male	1.000 (reference)	**1.581 (1.033–2.421)**	1.000 (reference)	**0.544 (0.310–0.957)**	1.000 (reference)	1.461 (0.953–2.239)	1.000 (reference)	0.866 (0.540–1.388)
Female	1.254 (0.769–2.044)	1.325 (0.823–2.134)	0.890 (0.618–1.281)	1.031 (0.586–1.812)	0.957 (0.601–1.525)	1.546 (0.945–2.531)	0.946 (0.641–1.394)	1.002 (0.606–1.657)
Hypertension (936)								
No	1.000 (reference)	1.060 (0.714–1.574)	1.000 (reference)	0.993 (0.598–1.650)	1.000 (reference)	1.415 (0.948–2.110)	1.000 (reference)	1.093 (0.715–1.672)
Yes	**2.085 (1.399–3.105)**	**3.120 (2.116–4.599)**	**2.811 (2.069–3.819)**	1.339 (0.808–2.218)	**2.725 (1.878–3.955)**	**3.256 (2.193–4.833)**	**2.720 (1.958–3.777)**	**2.000 (1.290–3.100)**
Diabetes mellitus (936)								
No	1.000 (reference)	1.154 (0.853–1.561)	1.000 (reference)	0.681 (0.454–1.020)	1.000 (reference)	1.184 (0.873–1.605)	1.000 (reference)	0.829 (0.596–1.154)
Yes	**2.019 (1.217–3.351)**	**2.904 (1.758–4.796)**	**2.048 (1.378–3.042)**	1.724 (0.833–3.567)	**1.788 (1.107–2.888)**	**2.984 (1.759–5.063)**	**1.779 (1.187–2.666)**	**2.883 (1.410–5.893)**
Hyperlipidemia (936)								
No	1.000 (reference)	1.180 (0.862–1.616)	1.000 (reference)	**0.594 (0.394–0.897)**	1.000 (reference)	1.340 (0.976–1.839)	1.000 (reference)	0.821 (0.582–1.159)
Yes	1.311 (0.834–2.059)	**1.769 (1.143–2.736)**	1.258 (0.894–1.771)	1.555 (0.770–3.140)	**1.601 (1.048–2.446)**	**1.739 (1.107–2.733)**	1.270 (0.881–1.832)	1.533 (0.891–2.637)
Smoking (930)								
No	1.000 (reference)	1.260 (0.900–1.764)	1.000 (reference)	0.799 (0.517–1.237)	1.000 (reference)	1.261 (0.899–1.769)	1.000 (reference)	0.989 (0.682–1.434)
Yes	1.478 (0.904–2.416)	**1.649 (1.033–2.632)**	1.216 (0.848–1.743)	0.880 (0.467–1.659)	1.212 (0.766–1.917)	**1.946 (1.172–3.229)**	1.303 (0.888–1.912)	1.095 (0.654–1.835)

NPRL3, nitrogen permease receptor like-3; MPG, N-methylpurine DNA glycosylase. *P*-values < 0.05 are shown in bold. The number of each subgroup is located next to each subgroup.

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
