# Peer review of "MPG and NPRL3 Polymorphisms Are Associated with Ischemic Stroke Susceptibility and Post-Stroke Mortality"

_diagnostics, 2020, doi:10.3390/diagnostics10110947_

Round 1
Reviewer 1 Report
Ischemic stroke is a complicated disease affected by environmental factors including genetic factors. Here, the various studies using whole-exome sequencing (WES) has focused on the novel and linkage variants in diverse diseases. So, authors investigated the various novel variants, which focused on linkage to each other, in ischemic stroke. Specifically, authors analyzed the N-methylpurine DNA glycosylase (MPG) gene, which plays an initiating role in DNA repair, and the nitrogen permease regulator-like 3 (NPRL3) gene, which is involved in regulating the mammalian target of rapamycin pathway. Authors showed that NPRL3 and MPG polymorphisms are associated with ischemic stroke prevalence and ischemic stroke survival.
Overall, this research is very interesting and yields important information for researchers and clinicians in this field. Therefore, NPRL3 and MPG genotypes would be a useful clinical biomarkers for for ischemic stroke development and prognosis. Overall, this paper has so many data and well-written.
Author Response
Response to the Reviewer’s comments
Reviewer 1
Ischemic stroke is a complicated disease affected by environmental factors including genetic factors. Here, the various studies using whole-exome sequencing (WES) has focused on the novel and linkage variants in diverse diseases. So, authors investigated the various novel variants, which focused on linkage to each other, in ischemic stroke. Specifically, authors analyzed the N-methylpurine DNA glycosylase (MPG) gene, which plays an initiating role in DNA repair, and the nitrogen permease regulator-like 3 (NPRL3) gene, which is involved in regulating the mammalian target of rapamycin pathway. Authors showed that NPRL3 and MPG polymorphisms are associated with ischemic stroke prevalence and ischemic stroke survival.
Overall, this research is very interesting and yields important information for researchers and clinicians in this field. Therefore, NPRL3 and MPG genotypes would be a useful clinical biomarkers for for ischemic stroke development and prognosis. Overall, this paper has so many data and well-written.
=>We thank the reviewer for the valuable comment and appreciate the reviewer's comment again.
Reviewer 2 Report
Authors have investigated the various novel variants, which focused on linkage to each other, in ischemic stroke. They analyzed the N-methylpurine DNA glycosylase (MPG) gene, which plays an initiating role in DNA repair, and the nitrogen permease regulator-like 3 (NPRL3) gene, which is involved in regulating the mammalian target of rapamycin pathway. Authors studies samples of 519 ischemic stroke patients and 417 controls. Genetic polymorphisms were detected by polymerase chain reaction (PCR), real-time PCR, and restriction fragment length polymorphism (RFLP) analysis. Authors found that two NPRL3 polymorphisms (rs2541618 C>T and rs75187722 G>A), as well as the MPG rs2562162 C>T polymorphism, were significantly associated with ischemic stroke. In Cox proportional hazard regression models, the MPG rs2562162 was associated with survival of small vessel disease patients in ischemic stroke. Authors concludes that NPRL3 and MPG polymorphisms are associated with ischemic stroke prevalence and ischemic stroke survival. Taken together, this suggests that NPRL3 and MPG genotypes may be useful clinical biomarkers for ischemic stroke development and prognosis.
This is a interesting and well written study
I have only minor comments to do :
1) did authors classify ischemic stroke patients according TOAST Classification ?
2) Did the authors analyzed the different frequence of each subtype of ischemic stroke in relation of the prevalence of each SNPs analyzed
3) authors on their discussion section should speculate bu some further sentence about the possible role of the studied SNPs on inflammatory actiation in the acute phase of stroke and they should add these citations on their reference section:
- Pinto A, Tuttolomondo A, Di Raimondo D, Fernandez P, Licata G. Risk factors profile and clinical outcome of ischemic stroke patients admitted in a Department of Internal Medicine and classified by TOAST classification. Int Angiol. 2006;25(3):261-267.
- Di Raimondo D, Tuttolomondo A, Buttà C, et al. Metabolic and anti-inflammatory effects of a home-based programme of aerobic physical exercise. Int J Clin Pract. 2013;67(12):1247-1253.
-Tuttolomondo A, Maida C, Pinto A. Diabetic foot syndrome as a possible cardiovascular marker in diabetic patients. J Diabetes Res. 2015;2015:268390
-Tuttolomondo A, Maida C, Pinto A. Diabetic foot syndrome: Immune-inflammatory features as possible cardiovascular markers in diabetes. World J Orthop. 2015 Jan 18;6(1):62-76.
Author Response
Response to the Reviewer’s comments
Reviewer 2
This is a interesting and well written study
I have only minor comments to do :
1) did authors classify ischemic stroke patients according TOAST Classification ?
=> We thank the reviewer for the valuable comment. As you mentioned, we have classified ischemic stroke patients according to the TOAST classification and modified the sentence in the material and methods. The modified sentence of the TOAST classification in the material and methods is as follows: “On the basis of the clinical symptoms and neuroimaging data, two neurologists categorized whole ischemic strokes into four causative subtypes using the criteria of the Trial of Org 10172 in Acute Stroke Treatment (TOAST) [38] as follows:”
2) Did the authors analyzed the different frequency of each subtype of ischemic stroke in relation of the prevalence of each SNPs analyzed
=> We thank the reviewer for the valuable comment. As you mentioned, we have analyzed the different group frequencies of each subtype of ischemic stroke about the prevalence of each SNPs analyzed.
The sentence of this response is as follows: “Furthermore, the results of a previous genome-wide association study (GWAS) studies showed that NPRL3 rs2541618 and MPG rs2562162 were associated with SVD patients and controls in the European population (Table S10).”
3) authors on their discussion section should speculate by some further sentence about the possible role of the studied SNPs on inflammatory activation in the acute phase of stroke and they should add these citations on their reference section:
=> We thank the reviewer for the valuable comment. As you mentioned, we added the sentences in the discussion part. The added sentences are as follows: “Although not these gene expressions studied in the ischemic stroke, the association between several risk factors of ischemic stroke, including inflammation activation and these gene polymorphisms, had speculated about the biomarker's possibility in the early diagnosis [50-53]. We thought it could be affected to the decision of ischemic stroke diagnosis in the combined high-risk factor levels and these polymorphisms.”
The added references are as follows:
- Pinto, A.; Tuttolomondo, A.; Di, Raimondo, D.; Fernandez, P.; Licata, G., Risk factors profile and clinical outcome of ischemic stroke patients admitted in a Department of Internal Medicine and classified by TOAST classification. Int Angiol 2006, 25, (3), 261-67.
- Di, Raimondo, D.; Tuttolomondo, A.; Buttà, C.; Casuccio, A.; Giarrusso, L.; Miceli, G.; Licata, G.; Pinto, A., Metabolic and anti-inflammatory effects of a home-based programme of aerobic physical exercise. Int J Clin Pract 2013, 67, (12), 1247-53.
- Tuttolomondo, A.; Maida, C.; Pinto,, Diabetic foot syndrome as a possible cardiovascular marker in diabetic patients. J Diabetes Res 2015, 268390
- Tuttolomondo, A.; Maida, C.; Pinto,, Diabetic foot syndrome: Immune-inflammatory features as possible cardiovascular markers in diabetes. World J Orthop 2015, 6, (1), 62-76.
Reviewer 3 Report
The authors investigated the association of MPG and NPRL3 SNP with ischemic stroke susceptibility and post-stroke mortality. This study revealed that NPRL3 and MPG polymorphisms are associated with ischemic stroke prevalence and ischemic stroke survival. The manuscript is well written. The analysis the author did is straightforward. I have one comment, did the author consider the cardiovascular disease as a risk factor? For example, atrial fibrillation are definite causal risk factors for ischemic stroke. If the control group exclude the participants with cardiovascular disease?